# Kynurenic Acid Modulates the Expression of Genes and the Activity of Cellular Antioxidant Enzymes in the Hypothalamus and Hippocampus in Sheep

**DOI:** 10.3390/ijms25179428

**Published:** 2024-08-30

**Authors:** Tomasz Misztal, Katarzyna Roszkowicz-Ostrowska, Paweł Kowalczyk, Patrycja Młotkowska, Elżbieta Marciniak

**Affiliations:** The Kielanowski Institute of Animal Physiology and Nutrition, Polish Academy of Sciences, Instytucka 3 Str., 05-110 Jablonna, Poland; k.roszkowicz@ifzz.pl (K.R.-O.); p.kowalczyk@ifzz.pl (P.K.); p.mlotkowska@ifzz.pl (P.M.); e.marciniak@ifzz.pl (E.M.)

**Keywords:** kynurenic acid, cellular antioxidant enzymes, gene expression, hypothalamus, hippocampus, sheep

## Abstract

Kynurenic acid (KYNA), a tryptophan metabolite, is believed to exert neuromodulatory and neuroprotective effects in the brain. This study aimed to examine KYNA’s capacity to modify gene expression and the activity of cellular antioxidant enzymes in specific structures of the sheep brain. Anestrous sheep were infused intracerebroventricularly with two KYNA doses—lower (4 × 5 μg/60 μL/30 min, KYNA20) and higher (4 × 25 μg/60 μL/30 min, KYNA100)—at 30 min intervals. The abundance of superoxide dismutase 2 (SOD2), catalase (CAT), and glutathione peroxidase 1 (GPx1) mRNA, as well as enzyme activities, were determined in the medial–basal hypothalamus (MBH), the preoptic (POA) area of the hypothalamus, and in the hippocampal CA1 field. Both doses of KYNA caused a decrease (*p* < 0.01) in the expression of SOD2 and CAT mRNA in all structures examined compared to the control group (except for CAT in the POA at the KYNA100 dose). Furthermore, lower levels of SOD2 mRNA (*p* < 0.05) and CAT mRNA (*p* < 0.01) were found in the MBH and POA and in the POA and CA, respectively, in sheep administered with the KYNA20 dose. Different stimulatory effects on GPx1 mRNA expression were observed for both doses (*p* < 0.05-*p* < 0.01). KYNA exerted stimulatory but dose-dependent effects on SOD2, CAT, and GPx1 activities (*p* < 0.05-*p* < 0.001) in all brain tissues examined. The results indicate that KYNA may influence the level of oxidative stress in individual brain structures in sheep by modulating the expression of genes and the activity of at least SOD2, CAT, and GPx1. The present findings also expand the general knowledge about the potential neuroprotective properties of KYNA in the central nervous system.

## 1. Introduction

Kynurenic acid (KYNA) is one of the highly neuroactive products formed during the enzymatic transformation of tryptophan in the kynurenine metabolic pathway (KP) [1]. Within the central nervous system (CNS), KYNA acts as a non-selective antagonist of ionotropic receptors for excitatory amino acids, including glutamate N-methyl-D-aspartate (NMDA) and α7 nicotinic acetylcholine (α7nACh) receptors [1,2]. The compound is also synthesized in the peripheral organs and can be found in various dietary food products [3,4]. Although some results suggest that KYNA, which is synthesized or applied peripherally, may have central effects [5], it is believed that KYNA’s efficiency in penetrating the blood–brain barrier is low and its concentration in the CNS depends on local synthesis [6]. KYNA is produced in all types of cells present in the brain, including neurons, oligodendrocytes, and glial cells, with the involvement of kynurenine aminotransferases [7,8]. Fluctuations in KYNA concentrations have been observed in the mammalian brain during the pre- and postnatal periods, as well as in adulthood [9]. It is believed that high levels of KYNA in the fetal brain may play a specific role during neurodevelopment, particularly in antagonizing both NMDA and α7nACh receptors [10,11]. Metabolic alterations in KP during adulthood are usually associated with a broad spectrum of neurological and psychiatric disorders. Increased KYNA concentrations have been demonstrated in various brain structures, as well as in cerebrospinal fluid (CSF) in Alzheimer’s disease, schizophrenia, bipolar disorder, meningitis, autoimmune diseases, and inflammatory processes. Conversely, decreased levels of this compound have been found in Huntington’s disease, Parkinson’s disease, and multiple sclerosis [3]. Pharmacological treatments increasing brain KYNA concentration have been shown to reduce excitatory glutamatergic and cholinergic neurotransmission, whereas those capable of decreasing brain KYNA levels facilitate transmission and increase excitotoxic damage [12,13,14]. Modulating KYNA levels in the brain through pharmacological interventions also affects transmission activity within the dopaminergic and gamma-aminobutyric acid systems [15,16,17]. Consequently, considering the multifaceted action of KYNA, elucidating the mechanisms behind the neuroprotective effects of this compound poses a considerable challenge, often contingent on its concentration in the CNS.

Given the endogenous nature of KYNA and its association with various neuropathologies, the antioxidant activity of this compound has also been demonstrated. Several authors have found its effective ability to scavenge hydroxyl radicals in various non-biological experimental systems [18,19]. KYNA has also been shown to decrease the levels of important markers of oxidative damage produced by different pro-oxidants in tissue preparation [20]. The intensity of neurogenesis and the high sensitivity of the developing nervous system to reactive oxygen species (ROS) may explain the high levels of KYNA in the fetal brain [9]. CNS exposure to oxidative stress during adulthood is one of the factors contributing to neurological disorders [21]. However, KYNA is not a major antioxidant in the brain. Mammalian cells are well adapted to combat oxidative stress and neutralize ROS through the activity of antioxidant enzymes such as superoxide dismutase (SOD), catalase (CAT), and glutathione peroxidase (GPx) [22]. SOD is a group of enzymes that catalyze the dismutation of the highly reactive superoxide anion to O_2_ and the less reactive species hydrogen peroxide (H_2_O_2_). CAT decomposes H_2_O_2_ into water and molecular oxygen, while the main biochemical function of GPx is to reduce the concentrations of hydroperoxides derived from unsaturated fatty acids. The expression of antioxidant enzyme genes and their activity can be regulated by many factors, with cell oxidative status being a key determinant [23]. These processes are also affected by various endogenous molecules, such as cytokines, neurotrophins, and hormones [24,25,26]. Measuring the activity of these major antioxidant enzymes serves as a useful biomarker for the body’s exposure to oxidative stress [27]. Considering the multiple mechanisms relating to the neuroprotective effects of KYNA in the CNS [28], we hypothesized that they also include the activation of cellular antioxidant enzymes. Therefore, the aim of this study was to examine KYNA’s capacity to modify gene expression and the activity of antioxidant enzymes in specific structures of the sheep brain, such as the hypothalamus and hippocampus. Among the known isoforms of SOD and GPx, we selected SOD2 and GPx1 for our study because both isoforms of enzymes are characteristic of tissues rich in mitochondria, including those in the brain, and provide basic protection for neurons and astrocytes against oxidative stress [22]. Importantly, the sheep brain shares many structural similarities with the human brain and can be utilized in studies on mental disorders and neurodegenerative diseases [29].

## 2. Results

Expressions of gene transcripts for all antioxidant enzymes examined were found in both hypothalamic areas, the MBH and POA, as well as in the hippocampal CA1 field. Differences in the abundance of these transcripts among the treatment groups are shown for individual tissues in Figure 1A,C,E (for SOD2), Figure 2A,C,E (for CAT), and Figure 3A,C,E (for GPx1). Differences in the activity of the enzymes examined across the treatment groups are depicted for individual tissues in Figure 1B,D,F (for SOD2), Figure 2B,D,F (for CAT), and Figure 3B,D,F (for GPx1).

### 2.1. SOD2 Expression Levels and Activity

Both doses of KYNA caused a significant decrease (*p* < 0.01) in SOD2 mRNA expression in the MBH, POA, and CA1 field of the hippocampus compared to the control group. Furthermore, lower (*p* < 0.05) SOD2 transcript abundance was observed in the MBH and POA of sheep infused with the KYNA20 dose compared to those infused with the KYNA100 dose. On the other hand, sheep treated with different KYNA doses expressed SOD2 mRNA at similar levels in the hippocampal CA1 field (Figure 1A,C,E).

SOD2 enzymatic activity in hypothalamic and hippocampal tissues increased (*p* < 0.05 and *p* < 0.001) in response to the infusion of both KYNA doses compared to the control group. The increase was more pronounced in the MBH and POA after the infusion of the KYNA20 dose (Figure 1B,D,F).
Figure 1Relative mRNA abundance (mean ± SEM, **left panel**) and enzyme activity (U/min., mean ± SEM, **right panel**) of superoxide dismutase 2 (SOD) in the hypothalamic medial–basal area (**A** and **B**, respectively), the preoptic area (**C** and **D**, respectively), and in the hippocampal CA1 field (**E** and **F**, respectively) of sheep infused with control solution and the lower (total 20 µg, KYNA20) and higher (total 100 µg, KYNA100) doses of kynurenic acid (KYNA) in the third ventricle. Significance of differences: *, *p* < 0.05; **, *p* < 0.01; ***, *p* < 0.001.
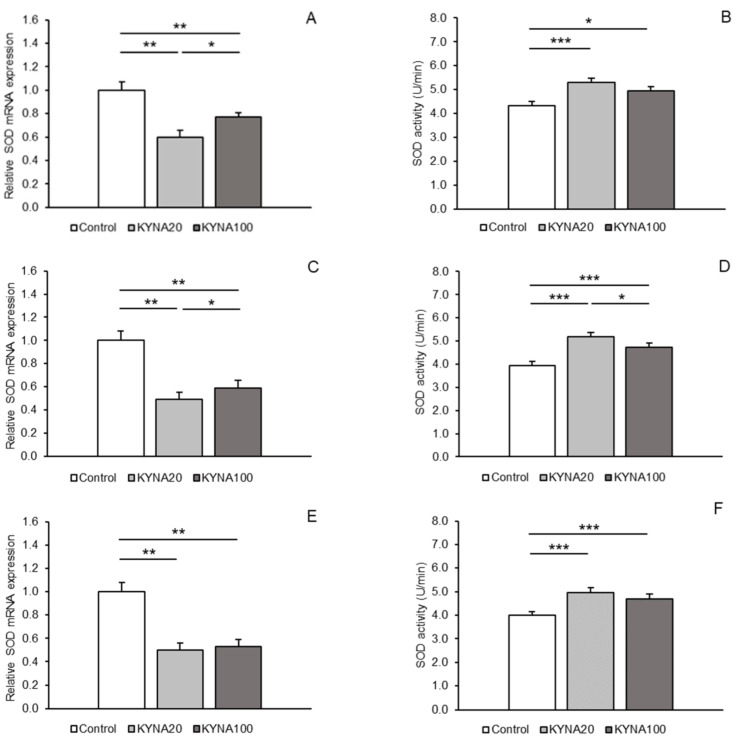



### 2.2. CAT Expression Levels and Activity

A consistent pattern in response to KYNA infusion was detected for CAT mRNA expression. Evident and highly significant (*p* < 0.01) decreases in expression were recorded in the MBH and hippocampal CA1 in response to both KYNA doses in comparison to the control group; however, CAT transcript abundance in the CA1 was lower (*p* < 0.05) in sheep infused with the KYNA20 dose than in those receiving the KYNA100 dose. In the POA, the KYNA20 dose reduced (*p* < 0.01) CAT mRNA levels, but the KYNA100 dose was ineffective compared to the control (Figure 2A,C,E).

For CAT, a progressive, dose-dependent increase (*p* < 0.001) in enzyme activity in the MBH was observed compared to the control group. However, in all tissues (MBH, POA and CA1), the highest enzymatic activity was recorded in the group infused with the KYNA100 dose (*p* < 0.001), compared to the other groups (Figure 2B,D,F).
Figure 2Relative mRNA abundance (mean ± SEM, **left panel**) and enzyme activity (U/min., mean ± SEM, **right panel**) of catalase (CAT) in the hypothalamic medial–basal area (**A** and **B**, respectively), the preoptic area (**C** and **D**, respectively), and in the hippocampal CA1 field (**E** and **F**, respectively) of sheep infused with control solution and the lower (total 20 µg, KYNA20) and higher (total 100 µg, KYNA100) doses of kynurenic acid (KYNA) in the third ventricle. Significance of differences: *, *p* < 0.05; **, *p* < 0.01; ***, *p* < 0.001.
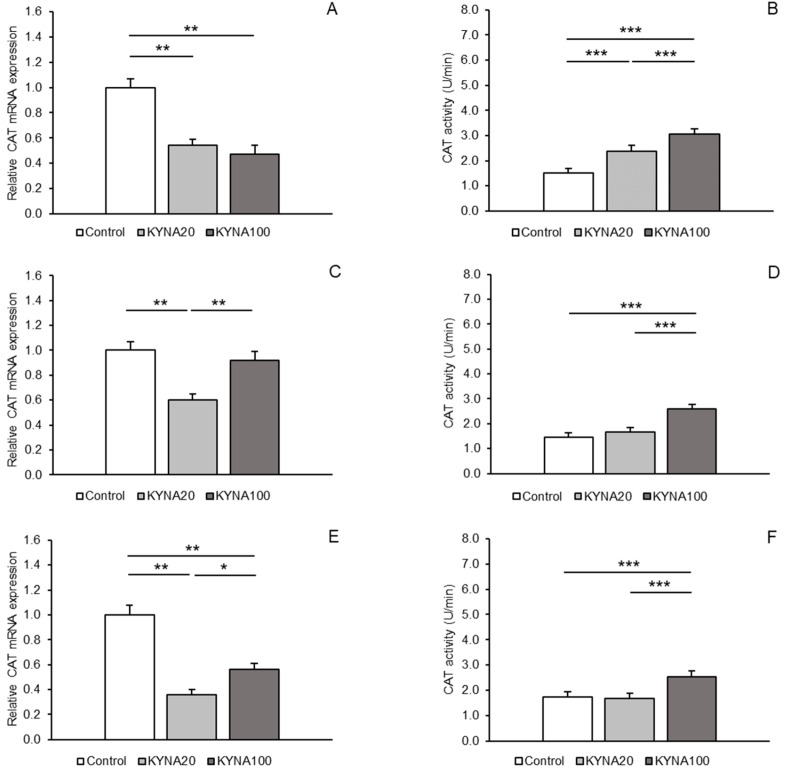



### 2.3. GPx1 Expression Level and Activity

Distinct stimulatory effects of the two KYNA doses were observed for GPx1 mRNA expression. While the KYNA20 dose increased (*p* < 0.01) GPx1 mRNA expression in the POA and hippocampal CA1 field, the KYNA100 dose elevated (*p* < 0.01) the expression in the MBH and CA1 compared to the control group. Moreover, a higher abundance (*p* < 0.05) of GPx1 transcript was noted in the CA1 field in sheep infused with the KYNA100 dose compared to those receiving the KYNA20 dose (Figure 3A,C,E).

The effect of KYNA administration on the enzymatic activity of GPx1 varied. The KYNA20 dose stimulated GPx1 activity in the POA and CA1 (*p* < 0.01 and *p* < 0.001), while the KYNA100 dose exerted a stimulatory effect in the MBH and also in the CA1 field (*p* < 0.001) compared to the control group. Additionally, the stimulatory effect of the KYNA100 dose was stronger (*p* < 0.01) in the CA1 field than that of the KYNA20 dose (Figure 3B,D,F).
Figure 3Relative mRNA abundance (mean ± SEM, **left panel**) and enzyme activity (µM/µg protein, mean ± SEM, **right panel**) of glutathione peroxidase 1 (GPx) in the hypothalamic medial–basal area (**A** and **B**, respectively), the preoptic area (**C** and **D**, respectively), and in the hippocampal CA1 field (**E** and **F**, respectively) of sheep infused with control solution and the lower (total 20 µg, KYNA20) and higher (total 100 µg, KYNA100) doses of kynurenic acid (KYNA) in the third ventricle. Significance of differences: *, *p* < 0.05; **, *p* < 0.01; ***, *p* < 0.001.
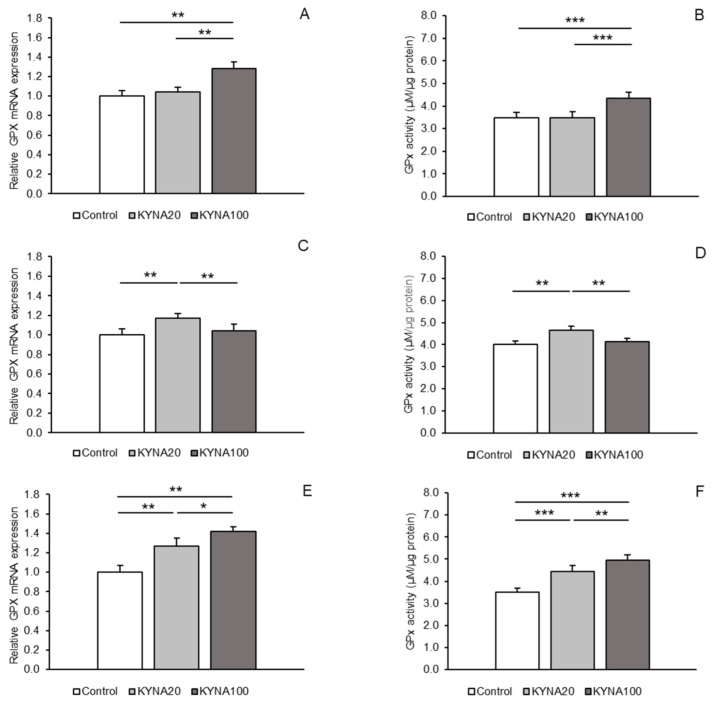


## 3. Discussion

The objective of this study was to analyze the effect of KYNA on the expression of genes and the activity of antioxidant enzymes in the hypothalamus and hippocampus in sheep under physiological conditions. These two structures, which in humans and other mammalian species consist of different anatomical areas and small neuronal nuclei, play crucial roles in regulating various physiological processes, such as metabolism and reproduction, as well as cognitive functions like learning, memory, and emotion. Furthermore, the hippocampus is recognized for sustaining physiological neurogenesis, which continues into adulthood across mammalian species, including humans [30]. Due to their high biochemical activity, these brain regions are continually exposed to highly reactive oxidants generated during normal cellular metabolism, particularly through the mitochondrial energy production pathway [31]. Many studies have demonstrated that an excessive accumulation of ROS in nerve cells can lead to oxidative stress, which in turn contributes to various neurological dysfunctions and neurodegenerative diseases [21,32]. Regarding the protection of CNS structures against the harmful effects of oxidants, we observed relatively high and comparable levels of SOD2, CAT, and GPx1 mRNA expression in the examined medial–basal and preoptic areas of the hypothalamus, as well as in the CA1 field of the hippocampus. What is more, KYNA appeared to be an important and effective modulator of this defense system: increasing brain KYNA levels by intracerebroventricular infusion resulted in significant and dose-dependent changes in gene expression and the activity of antioxidant enzymes in all structures examined. The potency of the observed response in the tissues may have been influenced by both the dose administered and the distance from the KYNA infusion site, located in the third ventricle. As described previously, the organization of CSF circulation in the sheep brain allows administered compounds to reach more distant areas, including, among others, the vicinity of the lateral ventricles [33]. Although the dose of infused KYNA appears to be higher than the physiological concentration described in the mammalian brain [28,34], a significant portion of KYNA was expected to flow out of the third ventricle into the mainstream and be absorbed into capillaries, e.g., into the median eminence or other circumventricular organs and the choroid plexus.

The expression of SOD2 and CAT gene transcripts decreased in response to KYNA administration in the MBH, POA, and CA1 field. These two enzymes are a very important pair in the cellular antioxidant defense system, converting the superoxide anion (O_2-_) into H_2_O_2_, and subsequently to water and oxygen [22,35]. The observed downregulation of mRNA expression in both antioxidant enzymes could suggest an adverse effect of the compound under study, leading to lower brain tissue protection against oxidative stress. However, considering KYNA’s ability to independently scavenge free radicals [18,19], it could be speculated that the increased brain KYNA levels resulted in a decreased demand for SOD2 and CAT in the cells. On the other hand, the natural decline in antioxidant enzymes in the body may be associated with age-related decreases in KYNA levels [9,36]. Interestingly, the decrease in SOD2 and CAT mRNA expression was in most cases stronger at the lower dose, indicating differences in sensitivity to KYNA between individual brain structures and/or a possible efficacy threshold for the doses applied. However, a more plausible explanation for the observed changes in the expression of SOD2 and CAT transcripts is increased translation, resulting in the higher synthesis of individual enzymes. This, in turn, was reflected by the enhancement of the efficiency of both antioxidant enzymes. While the enhancement of SOD2 activity was stronger in response to the lower dose, especially in the hypothalamic tissue, CAT activity was significantly more pronounced at the higher one in all tissues tested. Studies have shown that increasing tissue SOD level protects neurons against the death caused by oxidative injury and restores healthy mitochondrial morphology in monosodium glutamate-induced excitotoxicity disease models [37]. It is believed that impaired SOD2 is a potential pathogenesis related to oxidative stress in PD and AD [38], whereas the deficiency or malfunction of CAT is related to the pathogenesis of many age-associated degenerative diseases [39].

An entirely different pattern of response to KYNA was observed for GPx1 mRNA expression, where both the lower and higher dose exerted a stimulatory effect. Particularly, a marked gradual increase in GPx1 mRNA expression level was found in the CA1 field of the hippocampus. Furthermore, changes in the abundance of GPx1 mRNA in the tissues tested were reflected by the increased enzymatic efficiency of GPx1. This enzyme removes H_2_O_2_ by coupling its reduction with glutathione (GSH) oxidation and helps prevent lipid peroxidation, maintaining intracellular homeostasis and redox balance [22,35]. GSH is therefore an important endogenous compound, highly abundant in the brain, whose primary role is to neutralize free radicals in cells. The loss of GSH in hippocampal neurons has been shown to lead to dendrite disruption and cognitive impairment [40]. The research by Silva Ferreira et al. [41] demonstrated that KYNA administration in rats was able to prevent an increase in ROS production and the SOD/CAT ratio, as well as a decrease in GPx activity caused by another excitotoxic metabolite of tryptophan, i.e., quinolinic acid. More recently, the sub-chronic administration of L-kynurenine, an intermediate in KYNA synthesis, was found to elevate the GSH content and GPx activity in rat brain tissue, as well as to prevent oxidative damage induced by the ex vivo exposure of brain cells to pro-oxidants [42]. Additionally, a moderate elevation of brain KYNA levels was shown to reduce the histopathological and biochemical outcomes of experimentally induced ischemia [43]. Therefore, the observed increase in GPx1 synthesis and activity in our study may be associated with an increase in cellular GSH concentration and consequently a mobilization of antioxidant protection, particularly in the hippocampal CA1 field. Interestingly, increasing SOD levels in the body, e.g., through dietary supplementation, results in an increase in GPx activity in animal tissues [44]. Relationships between the activities of individual enzymes may also be visible in our study. Importantly, accumulating evidence indicates that GPx1 has multiple cellular functions: it is involved not only in the protection of cells against oxidative damage, but also in the regulation of metabolism and mitochondrial function, as well as in the control of cellular processes, such as apoptosis, growth, and signaling [45].

However, there is also evidence indicating harmful effects of KYNA on nerve cells, potentially leading to their damage [46,47]. It has been shown that the intrathecal infusion of KYNA into the spinal cord, lasting for several days, causes damage and a loss of myelin [46]. Moreover, rats infused with the highest concentrations of KYNA demonstrated adverse neurological signs, such as weakness and quadriplegia, linked to diffuse myelin damage. Another in vitro study [47] showed that KYNA administered at high levels reduced the viability of oligodendrocytes, and the mechanism of this action was distinct from that mediated by glutamatergic receptors. It is important to note that these deleterious effects of KYNA were associated with the long-term treatment of nervous tissue and extremely high concentrations of the compound applied. Considering the low nanomolar range of extracellular KYNA concentration in the mammalian brain, and the relatively fast turnover rate [28,34], the total concentration of the compound infused in our study could reach levels of several micromoles, predisposing it to interact with various types of receptors.

Since KYNA has several biological targets, understanding its neuroprotective effects is challenging, and multiple mechanisms have been proposed [48]. Apart from its well-established antagonistic effects on NMDA and α7nACh receptors [1,2], KYNA can activate the G-protein-coupled receptor 35 located in the CNS [49], with its potential relevance to the regulation of brain functions still under discussion [50]. KYNA is also an agonist for the aryl hydrocarbon receptor, an attractive target in neurodegenerative diseases, and may therefore contribute to immune and inflammatory regulation [48]. However, the effect of KYNA observed in our study could be largely related to the activation of the transcription factor Nrf2, which is responsible for the expression of a multitude of endogenous antioxidant agents [51]. As shown by the previously cited Silva Ferreira [41], the restoration of the activity of antioxidant enzymes by KYNA was associated with an increase in cytoplasmic and nuclear Nrf2 levels. Moreover, the ability of KYNA to alter cellular redox balance [18,19] may indicate mechanisms beyond known receptors. Therefore, a comprehensive understanding of the mechanism of KYNA’s antioxidant action in the CNS remains to be elucidated.

## 4. Materials and Methods

### 4.1. Animal Maintenance

Eighteen Polish Longwool sheep (a breed showing reproductive seasonality), aged 1 year and weighing 55 ± 2 kg, were used in the experiment. The animals were bred at the Sheep Breeding Center of the Kielanowski Institute of Animal Physiology and Nutrition, Polish Academy of Sciences (Jablonna near Warsaw, Poland), under natural lighting conditions (52° N, 21° E). They were fed twice daily according to their physiological status with a diet based on pellet concentrate, following the recommendations of the National Research Institute of Animal Production in Krakow-Balice (Poland) and the National Institute for Agricultural Research (France) [52]. During the experimental period, sheep were housed in individual pens, ensuring visual, olfactory, and tactile contact, and provided free access to water and mineral licks.

### 4.2. Third Ventricle Cannulation

One month before the experiment, the sheep underwent surgical implantation of a stainless-steel guide cannula into the third ventricle (IIIv) of the brain (outer diameter: 1.2 mm, frontal position: 31.0 mm). Implantation was performed under general anesthesia (xylazine: 40 mg/kg body mass, intravenously; xylapan and ketamine: 10–20 mg/kg body weight, intravenously; Bioketan, Vetoquinol Biowet, Pulawy, Poland), in accordance with the stereotaxic coordinate system for sheep hypothalamus [53] and the procedure described by Traczyk and Przekop [54]. The guide cannula was secured to the skull with stainless-steel screws and dental cement, and the external orifice of the canal was sealed with a stylet. After surgery, the sheep were injected for 4 days with antibiotics (1 g streptomycin and 1,200,000 IU benzylpenicillin; Polfa, Poland) and analgesics (metamizole sodium: 50 mg/animal; Biovetalgin, Biowet Drwalew, Poland, or meloxicam: 1.5 mg/animal; Metacam, Boehringer Ingelheim, Ingelheim am Rhein, Germany). The correct positioning of the cannula in the ventricle was confirmed for all sheep through the efflux of CSF during surgery and examination of the brain after slaughter.

### 4.3. Experimental Design and Tissue Collection

The experiment was performed in March during the natural anestrous season for this breed of sheep. The animals were randomly divided into three groups (*n* = 6 each) and infusion took place via the IIIv with Ringer-Locke solution (RLs, control) or with one of two KYNA doses (Sigma Chemical Co., St Louis, MO, USA) dissolved in RLs. The treatment was performed in a series of four 30 min infusions, at 30 min intervals, from 10:00 to 14:00. KYNA doses (lower, 4 × 5 μg/60 μL/30 min (KYNA20); higher, 4 × 25 μg/60 μL/30 min (KYNA100)) were selected on the basis of the scientific literature [28,34]. All infusions were performed using a BAS Bee microinjection pump (Bioanalytical Systems Inc., West Lafayette, IN, USA) and calibrated 1.0 mL gas-tight syringes. During the treatments, the sheep were kept in pairs in the experimental room in comfortable cages where they could lie down and to which they had been previously acclimated for three days. Directly after the experiment, the sheep were slaughtered after prior pharmacological stunning (xylazine 0.2 mg/kg body weight and ketamine 3 mg/kg body weight, intravenously), and the brains were promptly removed from the skull. Following the separation of the median eminence and cerebellum, each brain was sagittally dissected into cerebral hemispheres. The extracted blocks of the hypothalamus (cut to a depth of 2 mm) were dissected into two parts: the medial–basal hypothalamus (MBH) and the preoptic area (POA) [55], according to the ovine brain stereotaxic atlas [53]. The optic chiasm, thalamus, and mammillary body were utilized as landmarks. The hippocampus was dissected from the medial part of the temporal lobe of the right hemisphere, beginning from the floor of the lateral ventricle through the ventral and dorsal parts, according to the sheep brain atlas [56]. Approximately 2–3 mm sections were cut out from the CA1 field of the hippocampus. All tissue incisions were performed on sterile glass plates placed on ice, and subsequently the collected structures were immediately frozen in liquid nitrogen and stored at −80 °C.

### 4.4. Relative mRNA Abundance

Total RNA from the hypothalamic and hippocampal tissues was isolated using a NucleoSpin RNA II kit (Macherey-Nagel, Düren, Germany), according to the manufacturer’s protocol. The concentration and purity of isolated RNA were quantified using a NanoDrop ND-1000 spectrophotometer (Thermo Fisher Scientific, Waltham, MA, USA). RNA integrity was electrophoretically verified on 1.5% agarose gel stained with ethidium bromide. A TranScriba Kit (A&A Biotechnology, Gdynia, Poland) was used to synthesize cDNA according to the manufacturer’s instructions, utilizing 1 µg of total RNA in a reaction volume of 20 µL. Quantitative polymerase chain reaction (qPCR) was performed using 5× HOT FIREPol^®^ EvaGreen qPCR Mix Plus (Solis BioDyne, Tartu, Estonia). The PCR amplification mix contained 2 µL of cDNA template, 1 µL of primers (0.5 µL each at 10 pmol/mL), 3 µL of buffer PCR Master Mix, and 9 µL of dd H_2_O. The reaction conditions were as follows: initial denaturation at 95 °C for 15 min, denaturation at 95 °C for 15 s, annealing at 60 °C for 20 s, and elongation at 72 °C for 20 s (40 cycles). Specific primers for determining the expression of the *SOD2*, *CAT*, and *GPx1* genes, as well as endogenous control genes—glyceraldehyde-3-phosphate dehydrogenase (*GAPDH*) and peptidylprolyl isomerase C (*PPIC*)—were designed using Primer3 software (The Whitehead Institute, Boston, MA, USA) (Table 1). Amplification specificity was further validated by electrophoresis of the obtained amplicons in a 2% agarose gel and visualized under a UV light camera. Data were analyzed with Rotor Gene 6000 v. 1.7 software (Qiagen, Hilden, Germany) using a comparative quantification option and the Relative Expression Software Tool, based on the PCR efficiency correction algorithm developed by Pfaffl et al. [57,58]. The expression levels of the tested genes were normalized using geometrical means of the expressions of reference genes. Endogenous control genes were assayed in each sample to compensate for the variation in cDNA concentration and PCR efficiency between individual tubes.

### 4.5. Determination of Antioxidant Enzyme Activity

The total enzymatic activity of SOD was determined using the method described by Fridovich et al. [59], involving superoxide anion generation in the “xanthine–xanthine oxidase system” with NBT as the O_2_-detector. The principle of the method is based on two reactions. The first involves superoxide radical generation through the action of xanthine oxidase on xanthine. Subsequently, NBT reacts with the superoxide radical to form a navy-blue formazone dye, the incrementing level of which is recorded spectrophotometrically at 540 nm. A unit of SOD activity is defined as the amount of enzyme that reduces the reaction rate of NBT conversion to formazone by 50%, and the change in absorbance is 0.020 units/min.

CAT activity was determined spectrophotometrically by the method outlined by Aebi [60]. Triton X-100 solution was added to phosphate buffer to a final concentration of 0.1%. A unit of CAT activity is defined as the amount of enzyme required for the degradation of H_2_O_2_ within 2 min at 240 nm. 

The total enzymatic activity of GPx was determined using the method of Hopkins and Tudhope [61], involving the measurement of NADPH oxidation coupled with glutathione disulfide reduction by glutathione reductase. Changes in NADPH concentration were measured at 340 nm for 5 min. Enzyme activity was expressed as µM of oxidized NADPH_2_ per µg of protein. Therefore, the total protein concentration in tissue homogenates was analyzed spectrophotometrically using the Bradford method with the Bio-Rad Protein Assay Kit II (Bio-Rad, Hercules, CA, USA) according to the manufacturer’s instruction.

Determinations of all three enzymatic activities were performed using a Freedom EVO^®^ series pipetting station (Tecan, Mannedorf, Switzerland), a modular system with an incubator, shaker, and sample plate reader tailored to the specific application and required throughput. The Freedom Evo system is operated using Evo software, enabling control of the entire process carried out at the workstation.

### 4.6. Statistical Analysis

Initially, the Shapiro–Wilk test of normality was carried out to extract parametric and non-parametric data. Statistical assessments of differences in mRNA expression levels of SOD2, CAT, and GPx1 in the POA, MBH, and CA1 field between the treatment groups were carried out using non-parametric statistics, including the Kruskal–Wallis test with multiple comparisons of average ranks, and subsequently the Mann–Whitney *U* test for individual groups. Enzyme activity differences in the brain tissues tested between the groups were analyzed using one-way analysis of variance (STATISTICA, Stat Soft, Tulsa, OK, USA). Subsequently, each analysis was followed by the least significant difference post hoc test. Differences were considered significant at *p* < 0.05, and all data are presented as mean ± standard error of the mean (SEM).

## 5. Conclusions

In line with KYNA’s antioxidant activity, our study emphasizes the potential of this compound to modulate the expression of genes and the activity of cellular antioxidant enzymes (at least SOD2, CAT, and GPx1) in specific brain structures. Additionally, these results confirm the usefulness of the sheep model in studies concerning neurological processes in the CNS.

## Figures and Tables

**Table 1 ijms-25-09428-t001:** Sequences of primer pairs used in the study.

Gene	Primers (5′–3′)	Genbank Acc. No.	Amplicon Size
*SOD2*	F: GCAAGGAACAACAGGTCTTATCC R: ACTTGGTGTAAGGCTGACGG	NM_001280703.1	181
*CAT*	F: GAGCCCACCTGCAAAGTTCT R: CTCCTACTGGATTACCGGCG	XM_004016396.6	148
*GPx1*	F: TGTCGTACTCGGCTTCCC R: AGCGGATGCGCCTTCTCG	XM_004018462.1	163
*GAPDH*	F: GGGTCATCATCTCTGCACCT R: GGTCATAAGTCCCTCCACGA	NM_001190390.1	131
*PPIC*	F: TGGAAAAGTCGTGCCCAAGA R: TGCTTATACCACCAGTGCCA	XM_004008676.1	158

SOD2: superoxide dismutase-2, CAT: catalase, GPx1: glutathione peroxidase-1, GAPDH: glyceraldehyde-3-phosphate dehydrogenase, PPIC: peptidylprolyl isomerase C, F: forward primer, R: reverse primer. Real-time PCR amplification efficiencies of target and reference genes were 96–101%.

## Data Availability

The datasets analyzed during the current study are available from the corresponding author upon reasonable request.

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
