# Peer review of "Kynurenic Acid Modulates the Expression of Genes and the Activity of Cellular Antioxidant Enzymes in the Hypothalamus and Hippocampus in Sheep"

_ijms, 2024, doi:10.3390/ijms25179428_

Round 1

Reviewer 1 Report

Comments and Suggestions for Authors

The work done in this article appears to be quite thorough and detailed, there are however aspects in the study that should be improved or clarified.

The authors did not explain why they chose only 2 concentrations and did not do a dose effect.
What was the actual concentration of kynurenic acid in the CSF after injecting 20 or 100 ug of kynurenic acid?
How and in what was the kynurenic acid dissolved?
The figures and their description are not very clear, especially regarding relevance. They should be improved
.

The conclusions are controversial, particularly that  « … that KYNA plays a crucial role in maintaining redox homeostasis in brain cells ». Although the work appears technically sound, it does not appear to be based on a sufficiently thorough and detailed knowledge of the potential mechanisms of KYNA action in the brain. There are only indirect speculations about the action of KYNA.

Author Response

Response to the Reviewer 1

The authors thank very much for the critical comments regarding the manuscript. Referring to them, we tried to improve the content of the manuscript as much as possible.

Major points:

  1. The authors did not explain why they chose only 2 concentrations and did not do a dose effect.

Conducting research on large animal models is very expensive, especially if they involve complex surgical procedures. This requires additional maintenance of the animals for a period of 4 weeks for convalescence. The authors were not able to investigate the dose effect more extensively due to the low cost of the doctoral project. Such justification is not usually provided in the text. KYNA doses were selected on the basis of scientific literature and their possible distribution in the CSF was described in the Discussion (lines: 200-206).

  1. What was the actual concentration of kynurenic acid in the CSF after injecting 20 or 100 ug of kynurenic acid?

Measuring the actual KYNA concentration in the CSF would require a return CSF sample collection. This would be possible if the substance was administered into peripheral circulation, provided that it passed through the blood-brain barrier. Taking into account the active flow of CSF in the central nervous system, it can be assumed that the KYNA concentration changed dynamically, starting from the third ventricle, maintaining the level of several micrograms in further sites. This phenomenon is mentioned in the Discussion – lines: 200-206 and 268-271.

  1. How and in what was the kynurenic acid dissolved?

Kynurenic acid was dissolved in the Ringer-Locke solution - this is mentioned in the Materials and Methods section (line 318).

  1. The figures and their description are not very clear, especially regarding relevance. They should be improved.

The Results section has been reworded according to the suggestion of the Reviewer 2. The figures and their description have also been changed.

  1. The conclusions are controversial, particularly that  « … that KYNA plays a crucial role in maintaining redox homeostasis in brain cells ». Although the work appears technically sound, it does not appear to be based on a sufficiently thorough and detailed knowledge of the potential mechanisms of KYNA action in the brain. There are only indirect speculations about the action of KYNA.

The term related to redox homeostasis in brain cells has been removed. The authors are aware of the importance of determining the Nrf2 level in order to establish the detailed mechanism of action of KYNA in the CNS. Unfortunately, at this stage it is not possible due to the low cost of the doctoral project. Due to the lack of results regarding Nrf2, our conclusions have been slightly reworded:

“… In line with KYNA's antioxidant activity, our study emphasizes the potential of this compound to modulate the expression of genes and activity of cellular antioxidant enzymes (at least SOD2, CAT and GPx1) in specific brain structures. Additionally, these results confirm the usefulness of the sheep model in studies concerning the neurological processes in the CNS” – lines: 410-412.

Reviewer 2 Report

Comments and Suggestions for Authors

The manuscript presents the changes in mRNA expression and activity of basic antioxidant enzymes in the sheep brain after icv infusion of kynurenic acid. The experiments were performed on animals kept in conditions where the only factor that could change the physiological functioning of the brain was the mentioned KYNA infusion.

The manuscript has many shortcomings and most of them make it difficult to read and the presented results difficult to evaluate. Moreover, without additional data, like presentation of the Nrf2 reaction on KYNA infusion, the postulated conclusion that KYNA can modulate expression of antioxidant enzymes genes is only a speculation.

More detailed comments:

BBB permeability for KYNA is still debatable, the authors should take it into account in the Introduction and Discussion (Heyes MP, Quearry BJ., 1990; Scharfman HE, Goodman JH. 1998).

What is the aim of this study and what specific information can be obtained from it and how it can be used.

There is not information concerning Ethical Committee approval and compliance of procedures with EU directives.

Results are describe properly, but the data themselves are questionable. This concerns mostly enzyme activity. The units are unusual (fmol/µg protein/h). Can activity be presented as the effect of activity for an hour if, according to the Methods, measurements were carried out for up to 5 minutes? Moreover,  if within an hour 1 µg of protein transformed 4 fmol of the measured substance, the activity seems extremely low. Will there be any differences after sorting to standard units?

The markings on the figures are difficult to read. Statistical significance marks are not what is in the legend (Bc, Bd etc.); what is compared to what? Why the controls also have statistical marks? What is wrong with traditional asterix?

There is a noticeable lack of data on ROS; however, under the conditions tested, the level ROS and enzyme activity probably remained at a very low level. Testing the Nrf2 level would give strong support for the conclusions presented in this manuscript.

Comments on the Quality of English Language

Minor editing of English language required.

Author Response

Response to the Reviewer 2

The authors thank very much for the critical comments regarding the manuscript. Referring to them, we tried to improve the content of the manuscript as much as possible.

Major points:

  1. BBB permeability for KYNA is still debatable, the authors should take it into account in the Introduction and Discussion (Heyes MP, Quearry BJ., 1990; Scharfman HE, Goodman JH. 1998).

The citation by Scharfman and Goodman (1998), has been added in the Introduction [5] as the most relevant – lines: 37-39.

  1. What is the aim of this study and what specific information can be obtained from it and how it can be used.

The relevant information is included in the final part of the Introduction:

“Considering the multiple mechanisms relating to the neuroprotective effects of KYNA in the CNS [28], we hypothesized that they also include the activation of cellular antioxidant enzymes. Therefore, the aim of the study was to examine the KYNA’s capacity to modify gene expression and the activity of antioxidant enzymes in specific structures of the sheep brain, such as the hypothalamus and hippocampus. Among the known isoforms of SOD and GPx, we selected SOD2 and GPx1 for our study, because both isoforms of enzymes are characteristic of tissues rich in mitochondria, including the brain, and provide basic protection of neurons and astrocytes against oxidative stress [22]. Importantly, the sheep brain shares many structural similarities with the human brain and can be utilized in studies on mental disorders and neurodegenerative diseases [29]” – lines: 80-90.

  1. There is not information concerning Ethical Committee approval and compliance of procedures with EU directives.

The ethical statement is located after the main text, in accordance with the editorial requirements (line 420):

Statement of Ethical Matter: All animal procedures were conducted in accordance with the Polish Act on the Protection of Animals Used for Scientific or Educational Purposes (2015) and were approved by the 2nd Local Ethics Committee for Animal Experiments, Warsaw University of Life Sciences – SGGW, Warsaw, Poland (Resolution No. WAW2/128/2020) – lines 422-425.

  1. Results are describe properly, but the data themselves are questionable. This concerns mostly enzyme activity. The units are unusual (fmol/µg protein/h). Can activity be presented as the effect of activity for an hour if, according to the Methods, measurements were carried out for up to 5 minutes? Moreover, if within an hour 1 µg of protein transformed 4 fmol of the measured substance, the activity seems extremely low. Will there be any differences after sorting to standard units?

The authors would like to thank the Reviewer for pointing out the error, which concerned the enzyme activity. The activities of individual enzymes were measured using different methods and are expressed in separate units. The description of the methods has been reworded.

  1. The markings on the figures are difficult to read. Statistical significance marks are not what is in the legend (Bc, Bd etc.); what is compared to what? Why the controls also have statistical marks? What is wrong with traditional asterix?

The results have been presented in a new format and the readability of the figures has been improved.

  1. There is a noticeable lack of data on ROS; however, under the conditions tested, the level ROS and enzyme activity probably remained at a very low level. Testing the Nrf2 level would give strong support for the conclusions presented in this manuscript.

The authors are aware of the importance of determining the Nrf2 level in order to establish the detailed mechanism of action of KYNA in the CNS. Unfortunately, at this stage it is not possible due to the low cost of the doctoral project. Due to the lack of results regarding Nrf2, our conclusions have been slightly reworded: “In line with KYNA's antioxidant activity, our study emphasizes the potential of this compound to modulate the expression of genes and activity of cellular antioxidant enzymes (at least SOD2, CAT and GPx1) in specific brain structures. Additionally, these results confirm the usefulness of the sheep model in studies concerning the neurological processes in the CNS” – lines: 410-415.

Round 2

Reviewer 2 Report

Comments and Suggestions for Authors

The authors responded to the comments satisfactorily and introduced the suggested changes. The manuscript in its present form is suitable for publication.